# Differential DNA Methylation of THOR and *hTAPAS* in the Regulation of *hTERT* and the Diagnosis of Cancer

**DOI:** 10.3390/cancers14184384

**Published:** 2022-09-08

**Authors:** Pauline Ott, Marcos J. Araúzo-Bravo, Michèle J. Hoffmann, Cedric Poyet, Marcelo L. Bendhack, Simeon Santourlidis, Lars Erichsen

**Affiliations:** 1Epigenetics Core Laboratory, Institute of Transplantation Diagnostics and Cell Therapeutics, Medical Faculty, Heinrich-Heine University Duesseldorf, 40225 Duesseldorf, Germany; 2Group of Computational Biology and Systems Biomedicine, Biodonostia Health Research Institute, 20014 San Sebastián, Spain; 3IKERBASQUE, Basque Foundation for Science, 48013 Bilbao, Spain; 4Department of Urology, Medical Faculty, Heinrich-Heine University Duesseldorf, 40225 Duesseldorf, Germany; 5Department of Urology, University Hospital Zurich, 8091 Zurich, Switzerland; 6Department of Urology, University Hospital, Positivo University, Curitiba 80420-011, Brazil; 7Institute for Stem Cell Research and Regenerative Medicine, Medical Faculty, Heinrich-Heine University Duesseldorf, 40225 Duesseldorf, Germany

**Keywords:** DNA methylation, *hTERT*, urothelial cancer, THOR, *hTAPAS*

## Abstract

**Simple Summary:**

Because of its high prevalence of >45% in 9 out of 11 (82%) cancer types screened, THOR hypermethylation has been suggested to be a frequent telomerase-activating mechanism in *hTERT*-expressing tumor types, e.g., in cancers of the prostate, breast, blood, colon, lung, bladder, and brain. In this prime example, we present detailed DNA methylation profiles in urothelial cancer that reveal the exact positions of the most differentially methylated CpG dinucleotides within the THOR region in order to design an efficient Methylation-Specific PCR (MSPCR) approach for diagnostic and prognostic purposes. Furthermore, our data suggest an epigenetic mechanism regulating *hTERT* expression through the methylation status of THOR and lncRNA *hTAPAS.*

**Abstract:**

Background: Although DNA methylation in the gene promoters usually represses gene expression, the *TERT* hypermethylated oncological region (THOR) located 5′ of the *hTERT* gene is hypermethylated when *hTERT* is expressed in diverse cancer types, including urothelial cancer (UC). Methods: Comprehensive MeDIP and DNA methylation array analyses complemented by the technically independent method of bisulfite genomic sequencing were applied on pathologically reviewed and classified urothelial carcinoma specimens and healthy urothelial tissue samples to reveal the methylation status of THOR in detail. Results: The detailed DNA methylation profiles reveal the exact positions of differentially methylated CpG dinucleotides within THOR in urothelial cancer and provide evidence ofa diverging role of methylation of these CpGs in the regulation of *hTERT*. In particular, our data suggest a regulating mechanism in which THOR methylation acts on *hTERT* expression through epigenetic silencing of the lncRNA *hTERT* antisense promoter-associated (*hTAPAS*), which represses *hTERT*. Conclusions: These findings precisely define the most differentially methylated CpGs of THOR in early urothelial cancer, enabling optimal design of Methylation-Specific PCR (MSPCR) primers to reliably probe these methylation differences for diagnostic and prognostic purposes. In addition, this strategy presents a prime example that is also applicable to many other malignancies. Finally, the first evidence for the underlying epigenetic mechanism regulating *hTERT* expression through the methylation status of THOR is provided.

## 1. Introduction

Telomerase is an enzyme with major implications in human aging and disease through its role in establishing and controlling telomere length [1]. Telomerase functions as an RNA-dependent DNA polymerase [2] and consists of two essential components. The first component, the RNA *hTR* or *hTERC*, serves as a template for telomere synthesis by the second one, the catalytic protein *hTERT*, which possesses reverse transcriptase activity [3,4,5]. The RNA component is ubiquitously expressed in all human tissues, whereas *hTERT* is expressed only in specific cell types and aberrantly in cancer. In most somatic cells, the background transcription of *hTERT* is estimated to be less than 1 to 5 copies per cell, and an increase is correlated with telomerase activity [6]. The transcriptional activity of *hTR* alone is not sufficient for telomerase activity [7].

Telomeres are GT-rich repeats (TTAGGG)_n_ at the end of the chromosomes in humans and other vertebrates [8,9] that facilitate specific cellular functions. They serve as an essential barrier against carcinogenesis by protecting against chromosomal shortening, which would contribute to senescence [10,11]. In the absence of telomerase activity, telomeres shorten at every cell division because of the “end replication problem” [12,13]. When the telomere length has been critically shortened by continuous cell division or factors such as oxidative stress [14], cell proliferation is arrested and DNA damage responses as well as cellular senescence are triggered [15,16,17]. At this time, the cell reaches the so called “Hayflick limit” [18], also referred to as mortality stage 1 [19]. Cells may escape this status through the inactivation of several cell cycle regulators, which lead to further telomere loss and a second proliferative block and crisis, also referred to as mortality stage 2 [20,21]. Only a few cells are able to escape this state and maintain the residual telomere length, resulting in unlimited proliferative capacity or cellular immortality [22]. Most frequently, this escape is accomplished through the activation or upregulation of *hTERT*, which can be caused by several genetic disorders and can initiate and/or promote carcinogenesis [14]. Telomerase activity is observed in 85–90% of all human cancers, contributing to a limitless self-renewal capacity, although these malignant cells, for the most part, retain short telomeres [10,23]. Telomerase activation is commonly achieved through somatic mutations in the proximal promoter region of the *hTERT* gene, which are among the most common mutations in human cancers [14]. These can be observed in various types of cancer such as primary melanoma, glioblastoma, liposarcoma, and urothelial carcinoma [24,25]. However, there are also common tumor types, including breast, prostate, colorectal, and hematologic malignancies that have low mutation rates [25,26,27,28]. In addition, it has been observed that proviral insertions can have an impact on *hTERT* activation [29]. While these genetic changes are well studied, the epigenetic mechanisms controlling telomerase are thought to be critically important, but understudied [30].

However, some recent studies have indeed reported new epigenetic modes of *hTERT* activation. In this regard, one curious phenomenon has been described by different research groups for different tumor entities, which apparently contradicts the broadly established epigenetic paradigm that promoter methylation represses transcription [31,32]. Lee et al. reported an epigenetically regulated region upstream of the core promoter region of *hTERT* that appears to be responsible for its activation in cancer [32]. This region is referred to as the *TERT* hypermethylated oncological region (THOR) and was first described by Castelo-Branco et al. to be hypermethylated in *hTERT*-expressing pediatric tumors [33,34]. THOR is located within the *hTERT* promoter region (Chr 5: 1,295,321-1,295,753, GRCh37/hg19) and encompasses 433 bp with 52 CpG sites. Hypermethylation of this CpG-dense region has been shown to be responsible for upregulation of the *hTERT* expression in cancer, while the demethylation of THOR represses *hTERT* promoter activity [32]. Because of its high prevalence of >45% in 9 out of 11 (82%) cancer types screened, THOR hypermethylation has been suggested to be a frequent telomerase-activating mechanism in *hTERT*-expressing tumor types, e.g., in cancers of the prostate, breast, blood, colon, lung, bladder, and brain [32]. Importantly, in urothelial bladder cancer, where risk stratification remains an unsolved issue, hypermethylation of THOR is not only associated with a higher *hTERT* expression, but also with higher-risk disease in nonmuscle-invasive bladder cancers (pT1 high grade). Hypermethylation levels increased significantly in more severe cases [35]. Evidence for the prognostic value of increased THOR hypermethylation has also been provided for pancreatic cancer [36], glioblastoma, where THOR methylation is a dynamic process during glioma genesis [33], and for prostate cancer, where the possible risk stratification for low and intermediate Gleason cases by THOR hypermethylation has been suggested [34]. This is because THOR hypermethylation correlates with the Gleason scores and a connection with tumor invasiveness has been established. The authors of that study furthermore suggested that precise methylation mapping could provide additional valuable information for prognosis [34].

Another study, performed by Malhotra et al., reported a novel 1.6 kb non-coding RNA that is localized in the antisense direction to the *hTERT* promoter region and is capable of regulating *hTERT* expression [37]. This long non-coding RNA (lncRNA) is referred to as *hTERT* antisense promoter-associated (*hTAPAS*) RNA and its over expression is capable of down-regulating the *hTERT* transcript in trans [37]. *hTAPAS* transcription is initiated 167 nucleotides upstream of the *hTERT* transcription start site [37] (Figure 1A).

It is evident that the THOR region is located in the CpG-rich 5′ region of the *hTAPAS* transcript (Figure 1A). This prompted us to investigate the correlation between the methylation of this region and *hTAPAS* and *hTERT* gene expression. Our own observations as well as the general relevance of this region in cancer and in particular for stratification in bladder cancer, as proposed by Leão et al. [35], prompted us to carry out more detailed investigations. In these, we defined the DNA methylation profile of this region in pathologically examined, micro dissected tissue samples comprising primary pTa and pT1 tumors, as well as some advanced pT3a/pT3b tumors, and compared them to normal uroepithelial cells. The detailed DNA methylation profiles presented here provide evidence for differences regarding the importance of methylation at single CpG dinucleotides within a distinct epigenetically functioning regulatory region, and the corresponding expression data point to a new epigenetic mechanism that controls *hTERT* expression in early urothelial cancer. Furthermore, these findings set a prime paradigm presenting a basic strategy to optimize primers for Methylation-Specific PCR to reliably detect these methylation differences for diagnostic and prognostic purposes.

## 2. Material and Methods

### 2.1. Cell Lines and Cell Cultures

Cancer cell lines were obtained from ATCC and cultured in DMEM + Glutmax (Gibco, Waltham, MA, USA) supplemented with 10% FCS (Gibco) and 1% Penicillin and Streptomycin (Gibco) at 37 °C with 5% CO_2_. Cultivation of the primary uroepithelial cells was performed as described by Swiatkowski et al. [38].Primary Renal Mixed Epithelial Cells (ATCC, Manassas, VA, USA) were cultured as recommended by the manufacturer HREC (Normal Human Primary Renal Mixed Epithelial Cells, PCS-400-012, ATCC, Manassas, VA, USA), HBLAK [39], HEK293 [40], SW1710 [41], and HELA [42] were used.

### 2.2. Preparation of DNA from Formalin-Fixed, Paraffin-Embedded (FFPE) Tissue Samples and Cell Lines and Preparation of DNA Methylation Arrays

All of the methods were carried out in accordance with relevant guidelines and regulations. We confirm that the experimental protocols were approved and informed consent was obtained from all participants. An appropriate ethics vote was granted by the Kantonale Ethikkommission Zürich in 22.02.2013, Ref. Nr. KEK-ZH-Nr. 2012-0352. All details regarding the preparation of DNA from formalin-fixed, paraffin-embedded (FFPE) tissue samples, and the preparation of DNA methylation arrays have been described in a previous study, Erichsen et al., 2018 [43]. In this study, we also presented a table with all details on the clinical samples that were used.

In brief, we used four reference specimens of healthy urothelium, compared with ten UC tissue specimens from unifocal tumors (UT); four specimens adjacent to a unifocal tumor, histologically verified as benign, normal urothelial tissue (adj UT); five UC tissue specimens from multifocal tumors (MT); and five specimens adjacent to a multifocal tumor, histologically verified as benign, normal urothelial tissue (adj MT); Figure 2A. Statistical significance of the detected differential methylation in the tumors and tumor adjacent tissue groups was evaluated using two sample Student’s *t*-tests.

### 2.3. RNA Preparation, cDNA Synthesis, and Real-Time PCR

RNA was prepared using the RNeasy Mini Kit (Qiagen, Hilden, Germany) according to the manufacturer’s instructions. First-strand cDNA synthesis was performed from 1 μg RNA using the Super Script IV, First Strand Synthesis System (Invitrogen™). Primer annealing was achieved by incubating RNA, Oligod T primer, and dNTPs for 5 min at 65 °C, followed by 1 min incubation on ice. After adding the reaction mixture, including the transcriptase and RNase inhibitor, the reaction was incubated at 52 °C for 10 min.

Real-time PCR was carried out with SYBR Green PCR Master mix (Applied Biosystems, Foster City, CA, USA) using 10 ng template cDNA. All reactions were run in triplicate on a Step One Plus System (Applied Biosystems, v2.1, Waltham, MA, USA). Standard curves were generated using Step One software v2.1 (Applied Biosystems). Relative changes in the gene expression were calculated following the ΔΔ Ct-method with beta-Actin (*ß-ACTIN*) mRNA as a standard. Statistical significance of the detected differential transcription was evaluated using two sample Student’s *t*-tests with a level of significance *p* < 0.05. Primers were designed after excluding pseudogenes or other closely related genomic sequences that could interfere with specific amplification by amplicon and primer sequences comparison in the BLAT sequence database (https://genome.ucsc.edu/FAQ/FAQblat.html, accessed on 20 February 2022).

The following primers were used:

*hTERT* transcript s: 5′-CGGAAGAGTGTCTGGAGCAA-3′

*hTERT* transcript as: 5′-GGATGAAGCGGAGTCTGGA-3′

TM: 56 °C, Product length: 150 bp

*hTAPAS* transcript s: 5′-TGTAGCTGAGGTCGGCAAAC-3′

*hTAPAS* transcript as: 5′-GGTGCGAGGCCTGTTCAAAT-3′

TM: 60 °C, Product length: 128 bp

*ß-ACTIN* transcript s: 5′-GCCCAGTCCTCTCCCAAGTCCACACAG-3′

*ß-ACTIN* transcript as: 5′-GGGGGGGCACGAAGGCTCATCATT–3′

TM: 60 °C, Product length: 142 bp

The amplification conditions were denaturation at 95 °C for 10 min, followed by 40 cycles of 95 °C for 30 s, TM for 40 s, and 72 °C for 25 s.

### 2.4. Bisulfite Genomic Sequencing

Bisulfite sequencing was performed following bisulfite conversion with the EpiTec Kit (Qiagen, Hilden, Germany), as described previously [44,45]. The amplification conditions were denaturation at 95 °C for 13 min, followed by 35 cycles of 95 °C for 50 s, TM for 40 s, and 72 °C for 30 s. We used 10–20 ng of bisulfite converted DNA, 1 µL of a 200 μM dNTP Mix solution (Promega, Madison, WI, USA), 10 pM of each primer, and 10.5 units of Hot Star Taq DNA Polymerase (Qiagen) per reaction. The TA Cloning Kit (Invitrogen, Carlsbad, CA, USA) was used for cloning of the amplification products according to the manufacturer’s instructions. Sequence evaluation was performed with the Big Dye Terminator Cycle Sequencing Kit (Applied Biosystems) on a DNA analyzer 3700 (Applied Biosystems) using the M13-as primer. On average, 20 clones were sequenced to obtain the methylation profile of one sample. All of the sequences were aligned using CLUSTLW from the Kyoto University Bioinformatics Center on http://www.genome.jp/tools/clustalw/, accessed on 20 April 2022, and all methylated CpGs were manually counted for every single CpG position. PCR primers for the specific amplification of *hTERT* promoter sequences are as follows:

*hTERT* bseq s1: 5′-GGGTTTGTGTTAAGGAGTTTA-3′

*hTERT* bseq as1: 5′-ACCATAATATAAAAACCCTAA-3′

TM: 52 °C, Product length: 268 bp

*hTERT* bseq s2: 5′-GGTTTGTGTTAAGGAGTTTAAGT-3′

*hTERT* bseq as2: 5′-ATATAAAAACCCTAAAAACAAATAC-3′

TM: 52 °C, Product length: 262 bp

MRNA and 5′-regulatory gene sequences refer to the following sequences:

Homo sapiens telomerase reverse transcriptase (*TERT*), transcript variant 1, mRNA and Homo sapiens telomerase reverse transcriptase (*TERT*) gene, complete NCBI Reference Sequence: NM_198253.3 and AH007699.2

### 2.5. Relative Quantification of THOR Methylation

The methylation state of the partial THOR region (between −501 and −640 relative to the translation start of *hTERT*) was analyzed using Idiolocal Normalized Methylation-Specific PCR (IDLN-MSPCR) [46]. Purified plasmids containing the converted sequence of this region, with a known methylation state, were measured. For normalization primer without CpGs, in this case, primers *hTERT* bseq s1/as2 were used. In case of primary urothelial carcinoma, the *hTERT* CpGless s/as primers were used.

*hTERT* ms s:5′-TTCGGGAGGTTTCGCGTGTTC-3′

*hTERT* ms as:5′-TCCAAATCCGAACGCGAAACG-3′

TM: 56 °C, Product length: 139 bp

*hTERT* CpGless s:5′-GATTTGGGGTGGTTTGTTTATG-3′

*hTERT* CpGless as:5′-AATACCTCCCTACAACACTTCCC-3′

TM: 56 °C, Product length: 116 bp

Amplification was carried out using a Step One Plus Real Time PCR System (Applied Biosystems, Foster City, CA, USA), the Power SYBR Green PCR Master Mix (Applied Biosystems, Foster City, CA, USA), and the same amplification program as described for the real time PCR. Methylation levels were calculated using the ΔΔCT method.

## 3. Results

### 3.1. DNA Methylation of the CpG Island Direct Upstream of the HTERT Core Promoter Region Correlates with hTAPASRepression and HTERTExpression

The *hTERT* regulatory region THOR comprises the CpG-rich 5′ region of the *hTERT* antisense promoter-associated (*hTAPAS*) RNA gene (Figure 1A). Therefore, we first decided to replicate the reported correlation between *hTAPAS* RNA and *hTERT* transcription in different *hTERT*-expressing and non-expressing cell types as a prerequisite to subsequently analyze the underlying DNA methylation patterns of THOR in these cells.

We found that primary urothelium (pU), primary renal epithelial cells (hREC), and the benign urothelial cell line HBLAK do not express *hTERT* at high levels, but show an increased *hTAPAS* expression through direct comparison to the adenovirus type 5 (Ad5) immortalized embryonic human kidney cell line HEK293, the urothelial carcinoma cell line SW1710, and the cervix carcinoma cell line HeLa. These three cell lines all had a diminished *hTAPAS* expression, but increased *hTERT* mRNA expression (Figure 1B). The respective DNA methylation profiles showed comparatively low methylation of the THOR region, namely 10% in pU, 29% in hREC and 49% in HBLAK. In the latter two cell lines, methylation followed an unusual pattern in which certain CpGs were methylated while others were spared methylation (Figure 1C). In comparison, HEK293, SW1710, and HeLa showed dense DNA methylation of 99, 96, and 85% of this region (Figure 1C). Thus, DNA methylation correlates with *hTAPAS* repression and *hTERT* expression in these cell types.

### 3.2. Hypermethylation Constitutes Distinct DNA Methylation Profiles of the THOR/hTAPAS CpG-Island Direct Upstream of the hTERT Core Promoter Region in Early Urothelial Cancer

Genome-wide DNA methylation datasets were generated from pathologically reviewed and classified urothelial carcinoma and healthy urothelial tissue specimens by MeDIP and array analyses, as described in Erichsen et al. [43,47]. In brief, in this study, all other groups were compared to four samples of healthy urothelial tissue (pU). The other four groups of tissue samples consisted of ten specimens of UC from unifocal papillary tumors (UT); four specimens adjacent to unifocal tumors, histologically verified as normal urothelium (adj UT); five tissue specimens of UC from multifocal tumors (MT), and five specimens adjacent to multifocal tumors, histologically verified as normal urothelium (adjMT). For each individual sample, integrated peak values for overall methylation of the relevant CpG-rich region direct upstream of the *hTERT* core promoter was obtained. The mean values of these peak values were calculated for each sample group. The statistical significance of differentially methylated regions (DMRs) was calculated for each group of samples compared with the healthy urothelium reference, based on the mean peak values using two sample Student’s *t*-tests with a significance threshold of αDMR = 0.05. The differences in the mean values are given in Figure 2A. Within the 5′-regulatory region of *hTERT,* a significant hypermethylation was observed in MT, adj MT, and UT, but not in adj UT (Figure 2A).

We then established the detailed DNA methylation patterns in 11 pTa and pT1 urothelial bladder cancer specimens by bisulfite sequencing. In Figure 2B, we provide comprehensive DNA methylation profiles of the THOR region from four pTa low grade (LG), one pTa high grade (HG), and six pT1HG urothelial cancer samples. These profiles reveal a distinctive DNA methylation pattern in early urothelial cancers constituting of specific CpG dinucleotides, which are consistently methylated, while others remain largely unmethylated. In each of the four pTaLG samples, one sequence was methylated thoroughly, while in three of the pT1HG samples, the overall methylation level was increased as a result of two to five completely methylated sequences. In contrast, the same region was weakly methylated in the primary urothelial samples (Figure 1C and Figure 2B). Thus, hypermethylation of this CpG rich region nearby the *hTERT* core promoter and inside the THOR sequence and the 5′-*hTAPAS* region occurs in early stage urothelial cancer. A characteristic pattern of partial methylation predominates in pTa carcinomas, which was preserved in pT1 tumors. In some of the latter, however, more densely methylated sequences could also be observed.

### 3.3. Dense DNA Methylation of the THOR/hTAPAS CpG-Island in Advanced Stage Urothelial Cancer and the Correlation of hTAPAS Repression with the HTERT Expression in Primary Urothelial Cancer Cells of Diverse Stagesand Grades

Next, we analyzed in detail the DNA methylation patterns of one pT3a and two pT3b micro dissected tumor tissues together with one pT3b adjacent benign tissue. The data in Figure 3A reveal that in two cases, the methylation profiles consist, in part, of the already known characteristic pattern with methylation affecting certain CpGs, while sparing others, and, in part, of completely methylated sequences. Interestingly, one pT3b sample, in contrast with all other cancer tissue samples, exhibited almost complete methylated sequences (96% methylation), while the corresponding adjacent tissue showed the known partially methylated profile (Figure 3A).

Subsequently, we measured the expression of *hTAPAS* and *hTERT* in one pTaLG, one pT2, one pT3a, and one pT3b in comparison with two primary uroepithelial tissue samples and found an inverse correlation, but less pronounced, in the pT3b sample (Figure 3B).

### 3.4. Discrimination between Hypo, Partial, and Dense Methylation of THOR by IdiolocalNormalized Methylation Specific PCR (IDLN-MSPCR)

Finally, based on our data on THOR-associated differential methylated CpG patterns in urothelial cancer, we designed primers to distinguish between hypo, partial, and dense methylated CpG positions within this THOR region. As templates, we chose plasmids that after sequencing, were verified to bear the intended THOR sequences of differential methylation (Figure 4). Six plasmids with hypomethylated THOR inserts, eight plasmids with partially methylated inserts, and seven plasmids with enhanced methylated inserts were used (Figure 4A). In Figure 4B, it is documented that in this relatively quantifying EpiTHOR *hTERT* assay, the amplification efficiency of the hypomethylated sequences was reduced by approximately 300-fold compared with that of the partially methylated sequences. This, in turn, was reduced approximately 70-fold compared with that of the densely methylated inserts. Thus, clear differentiation of these differential methylated genetic spots is feasible by this approach.

After this first proof of concept, the Epi THOR *hTERT* assay was also applied to primary urothelial carcinoma (Figure 5). A pTa, a pT2, a pT3a, and a pT3b were measured in comparison to a benign control (BN). The amplification efficiencies of the carcinomas were between 11- and 22-fold higher than that of the control, indicating higher methylation of the region analyzed in these tumors.

## 4. Discussion

In differentiated somatic cells, *hTERT* expression is absent. In contrast, upregulation of *hTERT* is observed in 85–90% of all human cancers, conferring a limitless self-renewal capacity, and is involved in tumor initiation and progression [14].

In recent studies, diverse new epigenetic modes of *hTERT* regulation have been reported [31,36]. In these studies, a differentially methylated region, named THOR, located slightly upstream of the *hTERT* core promoter region has been identified to be of regulatory importance for *hTERT* expression. Surprisingly, contrary to the central dogma of epigenetics, that the dense methylation of CpG-rich promoters is strictly correlated with transcriptional repression, in this case, methylation seems to increase hTERT expression. In addition, localized in the same region but being orientated in the opposed direction, a novel 1.6kb long non-coding RNA, named *hTAPAS*, which negatively regulates *hTERT* expression, has been reported by Malhotra et al. [37].

Taken together, our data document the proposed correlation between DNA methylation of the THOR region and the diminished and increased expression of *hTAPAS* and *hTERT*, respectively (Figure 1B,C). These findings suggest a new straightforward explanation for the apparent paradoxical relation between methylation of the THOR region and an increased *hTERT* expression, which stands in apparent contradiction to the classic assumptions about epigenetic repression by DNA methylation. According to this explanation, dense DNA methylation at the THOR region that is located in the 5′ region of *hTAPAS* would repress the expression of this long non-coding RNA. This would alleviate *hTERT* repression by *hTAPAS,* leading to increased *hTERT* expression in spite of higher methylation in the *hTERT* upstream region. Thus, the prime target of epigenetic repression by THOR methylation is *hTAPAS,* and the effects on *hTERT* expression are primarily indirect. Specifically, dense DNA methylation of the *hTAPAS* CpG-island appears to contribute to *hTERT* re-expression in cancer. Further studies should dissect the precise mechanism of this negative interference of the long non-coding RNA *hTAPAS* on *hTERT* expression, e.g., by RNA interference with the lncRNA *hTAPAS* in expressing and THOR unmethylated cell models, e.g., primary uroepithelium (Figure 1C and Figure 2A).

For urothelial carcinogenesis, our comprehensive DNA methylation pattern analysis moreover demonstrates increasing THOR methylation from normal urothelium through low-grade pTa and high grade pT1 to muscle-invasive cancers, in which methylation is dense, i.e., affecting essentially every CpG. At earlier stages, a partially methylated pattern is observed, which strongly suggests that not all CpGs are equivalent with respect to their methylation status and their role for *hTERT* expression regulation. In particular, our results from the cell lines suggest that a partially methylated pattern (found e.g., in HBLAK, Figure 1B,C) still allows for *hTAPAS* expression, whereas complete DNA methylation rather inhibits *hTAPAS* expression. In contrast, in some of the pT1HG samples, completely methylated sequences were found that predominate in muscle-invasive cancers. This finding may be explained by the selection process for the high methylation status at THOR, which favors *hTERT* expression and thereby the clonal expansion of cells that have escaped replicative senescence. Measurements of the relation between THOR DNA methylation and *hTERT* expression on a larger series, especially of pTaLG and pT1HG tumors, should be performed to further underline this idea.

Of note, it must always be considered that further differential methylated regions contribute to the tumor-associated aberrant expression of *hTERT*. For instance, a positive correlation between the hypermethylation status of 27 CpG sites within the *hTERT* promoter core region, *hTERT* mRNA expression, and telomerase activity in a comprehensive investigation of 56 human tumor cell lines, as well as tumor and normal tissues from different organs, have been shown [31]. Furthermore, the results of our study should not distract from the possible case that despite the full methylation of THOR, *hTAPAS* repression may be in some cases uncoupled from this, and the induction of *hTERT* expression may be the consequence of a general increase in CpG methylation within the entire *hTERT* promoter. This is indicated by the full methylated THOR and the expression of *hTAPAS* and *hTERT* of HEK293 (Figure 1B,C).

Here, we provide first evidence that the correlation between *hTAPAS* repression and *hTERT* expression exists not only in cell models (Figure 1B,C), but also in urothelial cancer tissue samples from different stages. It is noteworthy, that in the case of the dense methylated pT3b sample (Figure 3A), we found that the adjacent tissue carries the distinct partially methylated profile resembling that from pTa and pT1 tumors (Figure 2B). This observation further supports the notion of an epigenetic field effect during urothelial carcinogenesis affecting the morphologically normal-appearing tissue adjacent to the manifested tumors. This in turn emphasizes the cancer specific function of THOR methylation.

## 5. Conclusions

In summary, we conclude that the methylation profile of the THOR region mirrors the extent of the epigenetic impairment of *hTERT* expression control in early and advanced urothelial cancer. This may also hold true for many other cancer entities depending on *hTERT* activation, e.g., for prostate, breast, blood, colon, skin, and brain cancers [32]. Moreover, we suggest that detailed profiling of THOR methylation, as performed here, will be helpful to stratify cancers with respect to their aggressiveness. The methylation profiles, once detected, provide the basis for the design of primers for Methylation-Specific PCR (MSPCR) that will be able to effectively distinguish unmethylated from partially methylated and both from full methylated THOR regions. With this tool in our hand, we will for instance approach a main problem in urothelial cancer diagnosis to distinguish early relatively harmless pTa from life-threatening pT1HG tumors. Of course, follow-up and monitoring after therapy are also feasible by Epi THOR *hTERT* assay alone or in combination with other biomarkers, e.g., LINE-1 [43] or ODC1 [47], depending on the outcome of further validation studies. For this purpose, we will employ IDLN MSPCR, as presented here, which allows for reliable and fast detection of differential methylation in cancer samples with diversities in genetic aberrations [46]. In this context, we recently presented a simple but effective technical approach to separate cell-free DNA, known to be released from tumors in the surrounding tissues and body fluids, from cellular DNA. This method enables defining cancer-associated DNA methylation patterns of this THOR and other differential methylated regions from pure cell-free DNA extracted from blood, sputum, or urine. Here, we will unreservedly apply the above presented strategy to enable the early detection and follow-up of bladder cancer noninvasively from urinary samples. It has not escaped our notice that the presented approach may serve as part of a basic strategy for the establishment of epigenetic diagnostics and prognostics for many cancers from body fluids and tissue samples.

## Figures and Tables

**Figure 1 cancers-14-04384-f001:**
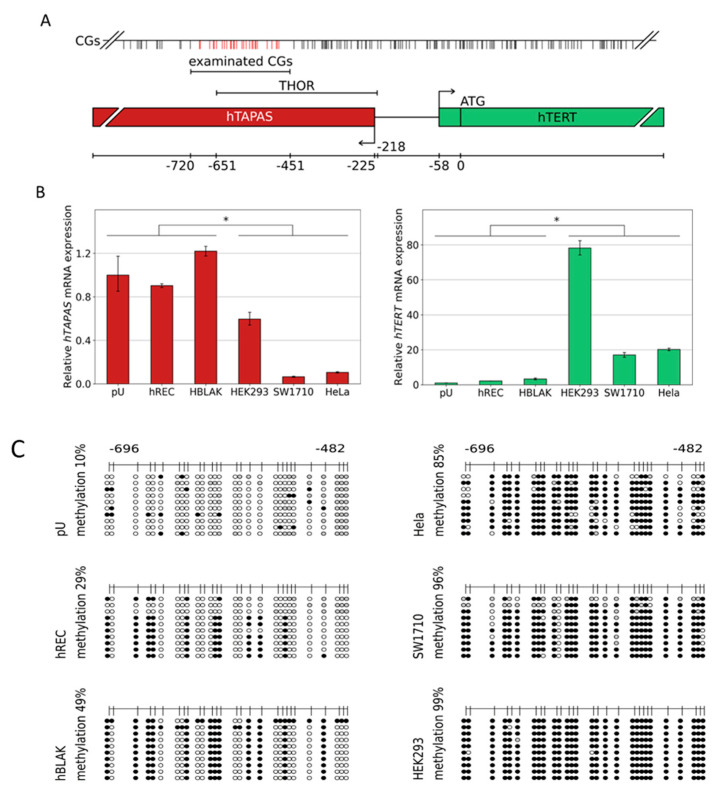
Relative *hTAPAS* and *hTERT* expression and THOR DNA methylation patterns in various cell types. The above-depicted illustration (**A**) provides the exact location of the examined CpG rich region and further details, e.g., the proportional distribution of the CpG dinucleotides within this genomic area. Primary urothelium and the cell lines hREC, HBLAK, HEK293, SW1710, and HELA were used to measure the expression of lncRNA *hTAPAS* and *HTERT* gene. * *p* ≤ 0.05 (**B**). In addition, the detailed DNA methylation patterns of a 215 bp 5′- fragment within THOR, ranging from −482 to −696 relative to the translational start site of *HTERT* gene, was determined from those cell types (**C**). Here, every unmethylated CpG dinucleotide is represented by a blank circle, every methylated one by a black circle, and non-defined CpG positions are depicted by grey circles.

**Figure 2 cancers-14-04384-f002:**
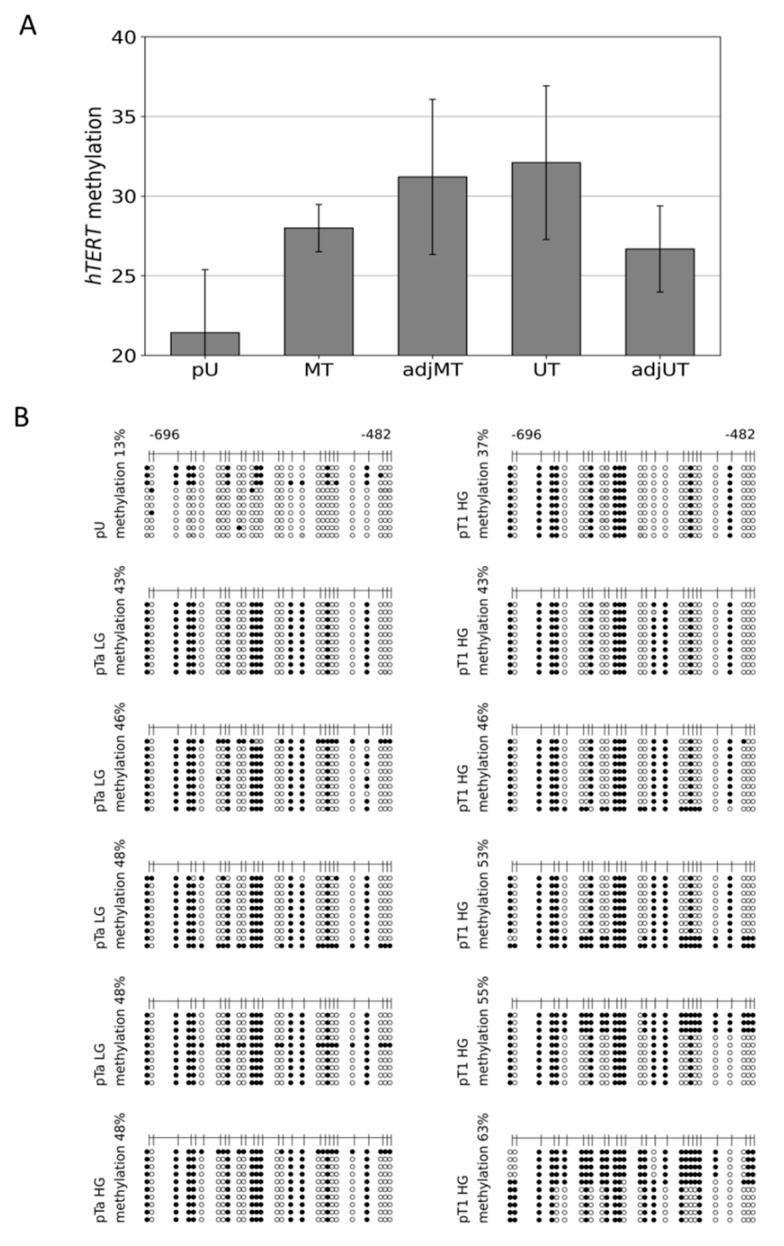
Mean peak values detected by microarray DNA methylation analysis of the THOR region at the 5′-region of the *hTERT* gene and detailed THOR methylation profiles of pTa LG/HG and pT1 HG urothelial cancer. Methylation grade by MeDIP/promoter array analyses: 4 primary urothelium (pU): 21.416, 5 multifocal tumors (MT): 27.992, 5 adjacent multifocal tumors (adj MT): 31.202, 10 unifocal tumors (UT): 32.099, and 4 adjacent unifocal tumors (adj UT): 26.671 (**A**). In addition, the detailed DNA methylation patterns of a 215 bp 5′-fragment within THOR, ranging from −482 to −696 relative to the translational start site of *HTERT* gene, were determined from 1 pU, 4 pTa LG, 1 pTa HG, and 6 pT1 HG urothelial cancer samples (**B**). Here, every unmethylated CpG dinucleotide is represented by a blank circle, every methylated one by a black circle, and non-determined CpG positions are depicted by grey circles.

**Figure 3 cancers-14-04384-f003:**
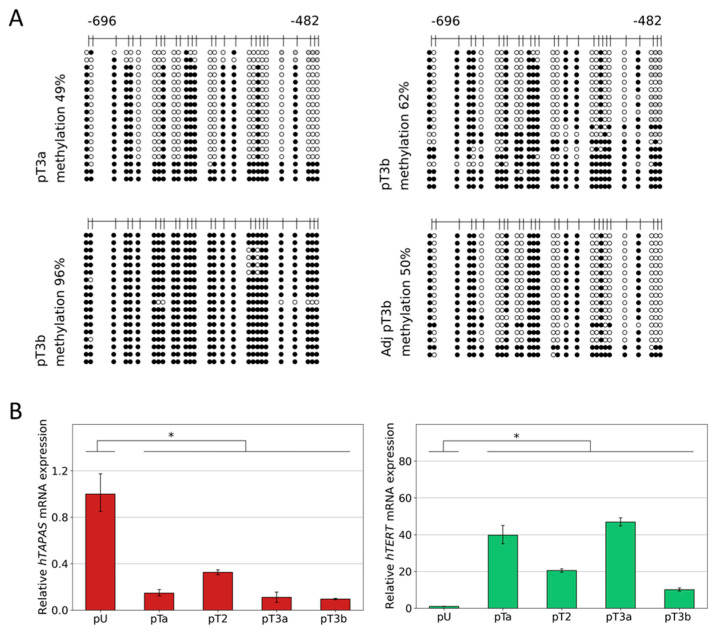
Relative *hTAPAS* and *hTERT* gene expression and detailed DNA methylation profiles of the THOR region in urothelial cancer. The detailed DNA methylation patterns of a 215 bp 5′-fragment within THOR, ranging from −482 to −696 relative to the translational start site of the *HTERT* gene, was determined from one pT3a, one pT3b, and one pT3b, and its adjacent benign urothelial tissue sample. Here, every unmethylated CpG dinucleotide is represented by a blank circle, every methylated one by a black circle, and non-defined CpG positions are depicted by grey circles (**A**). In addition, two samples of primary urothelium and samples of one pTaLG, one pT2, one pT3a (49% methylation), and one pT3b (96% methylation) were used to measure the expression of the lncRNA *hTAPAS* and *HTERT* genes, * *p* ≤ 0.05 (**B**).

**Figure 4 cancers-14-04384-f004:**
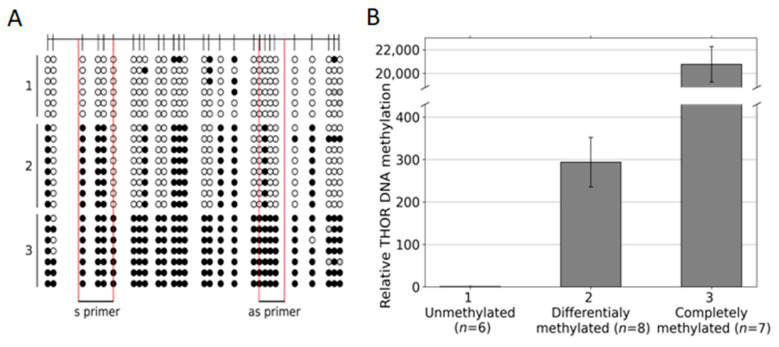
THOR Idiolocal Normalized Methylation-Specific PCR (IDLN-MSPCR) on hypomethylated, partially methylated, and full methylated THOR sequences. Six plasmid preparations with hypomethylated, eight with partially methylated, and seven with dense methylated THOR sequences (**A**) were measured by IDLN-MSPCR with the indicated primers (S1/AS1 TAPS) covering four CpG positions, each to define their relative DNA methylation status (**B**).

**Figure 5 cancers-14-04384-f005:**
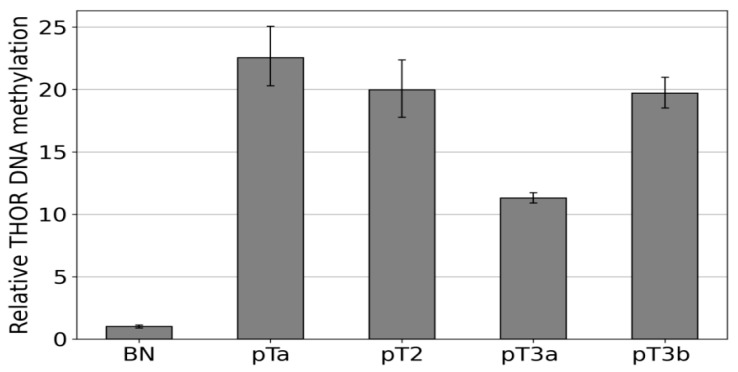
Methylation-specific amplification of the THOR region in urothelial carcinoma THOR methylation states were determined via IDLN-MSPCR in the following samples, pTa, pT2, pT3a, and pT3b. These were compared to a healthy control (BN).

## Data Availability

The data presented in this study are available.

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
