# Peer review of "Differential DNA Methylation of THOR and hTAPAS in the Regulation of hTERT and the Diagnosis of Cancer"

_cancers, 2022, doi:10.3390/cancers14184384_

Round 1
Reviewer 1 Report
In this manuscript, Ott et al. analyzed the differential DNA methylation of THOR and hTAPAS in the regulation of hTERT. They found that hTAPAS methylation and hTERT expression levels are correlated which means a high methylation in hTAPAS results in upregulation of hTERT which triggers carcinogenesis. Their study and findings are novel and important for CpG methylation pattern of THOR region within hTAPAS to be used as an diagnostic and prognosting marker especially in urothelial cancer which is difficult to stratify with respect to its aggressiveness. Another important point is the applicability of this marker system to other hTERT activity-mediated cancers such as prostate, breast, blood, colon, skin and brain cancers. Experimental design is appropriate to test their hypothesis.
However, there are some points to be reconsidered by the authors:
1- Overall, explanations and grammar should be checked and corrected throughout materials-methods section. Here are some examples:
-In materials- methods section, under title 2.1, the last sentence ‘ hREC, HBLAK,HEK293,SW1710, and HELA’ is incomplete.
-Under title 2.2, the last sentence is incomplete, again. In addition, inconsistent with the title, DNA preparation from formalin-fixed tissue and preparation of DNA methylation arrays are not described.
-Under title 2.3, it should be written as ‘primer annealing’ instead of ‘primer association’.
2- In figure 1, it would more accurate to sort the figures in the figure panel in the order in which they are explained in the figure caption.
3- Under title 2.2, while explaining experimental groups, they say 10 unilocal UT, and 4 benign adjacent tissue, but then, in 3.2 results section, they write it as ‘ 10 unilocal UT, and 5 benign adjacent tissue. It should be consistent throughout the text.
4- Also, the abbreviation for benign normal UT tissues is given as ‘badj UT’ in material-methods section, while it is given as ‘ adjUT’ in results section. To be consistent, this should be corrected.
5- Results are given in a very complicated way making it difficult to understand. A more clear presentation (especially in terms of length, complexity and structure of sentences) would be more acceptable.
Author Response
Dear reviewer, on behalf of all authors I thank you for reviewing our manuscript and your suggestions to improve it.
Comments and Suggestions for Authors
In this manuscript, Ott et al. analyzed the differential DNA methylation of THOR and hTAPAS in the regulation of hTERT. They found that hTAPAS methylation and hTERT expression levels are correlated which means a high methylation in hTAPAS results in upregulation of hTERT which triggers carcinogenesis. Their study and findings are novel and important for CpG methylation pattern of THOR region within hTAPAS to be used as an diagnostic and prognosting marker especially in urothelial cancer which is difficult to stratify with respect to its aggressiveness. Another important point is the applicability of this marker system to other hTERT activity-mediated cancers such as prostate, breast, blood, colon, skin and brain cancers. Experimental design is appropriate to test their hypothesis.
However, there are some points to be reconsidered by the authors:
1- Overall, explanations and grammar should be checked and corrected throughout materials-methods section. Here are some examples:
-In materials- methods section, under title 2.1, the last sentence ‘ hREC, HBLAK,HEK293,SW1710, and HELA’ is incomplete.
Sentence has been completed
-Under title 2.2, the last sentence is incomplete, again. In addition, inconsistent with the title, DNA preparation from formalin-fixed tissue and preparation of DNA methylation arrays are not described.
Sentence has been completed and the reference including all details has been added.
-Under title 2.3, it should be written as ‘primer annealing’ instead of ‘primer association’.
Has been changed
2- In figure 1, it would more accurate to sort the figures in the figure panel in the order in which they are explained in the figure caption.
Have been sorted
3- Under title 2.2, while explaining experimental groups, they say 10 unilocal UT, and 4 benign adjacent tissue, but then, in 3.2 results section, they write it as ‘ 10 unilocal UT, and 5 benign adjacent tissue. It should be consistent throughout the text.
Has been made consistent
4- Also, the abbreviation for benign normal UT tissues is given as ‘badj UT’ in material-methods section, while it is given as ‘ adjUT’ in results section. To be consistent, this should be corrected.
Have been corrected
5- Results are given in a very complicated way making it difficult to understand. A more clear presentation (especially in terms of length, complexity and structure of sentences) would be more acceptable.
Has been addressed
Reviewer 2 Report
Generasl comments
In the manuscript by Ott et al, the authors evaluated DNA methylation profiles of the THOR region of the hTERT promoter in various cell lines, as well as in early noninvasive urothelial cancers and muscle-invasive cancers. Regarding cell lines, the data show increased methylation of THOR at specific CpGs from the primary urothelium (pU) to primary renal epithelial cells (hRECs) and the benign urothelial cell line HBLAK. Furthermore, the authors show that methylation further extended to multiple adjacent CpGs in Hela cells and in the SW1710 cell line, while each CpG was methylated in HEK293 cells. Interestingly, analyses of the mRNA expression of hTERT and the long noncoding RNA of hTAPAS in these cell lines suggest that the specific pattern of CpG methylation detected in hREC and HBLAK cells is associated with high expression of hTAPAS mRNA but low expression of hTERT mRNA, corroborating previous studies showing the ability of hTAPAS to trans-repress hTERT mRNA. On the other hand, the expression levels of hTAPAS/hTERT in HEK293 cells suggest that the fully methylated state of THOR in this cell line may be uncoupled from hTAPAS repression and the induction of hTERT expression may be the consequence of a general increase in CpG methylation within the entire hTERT promoter (see below for further criticism).
As regard urothelial cancer, the specific CpG methylation pattern detected in HBLAK cells was also present in both low-grade and high-grade pTa cancers, whereas in high-grade pT1 muscle-invasive cancer methylation affected every CpG ( as in HEK293 cells).
Similar pattern of THOR methylation were also shown for pT3a and two pT3b micro-dissected tumor tissues and one pT3b adjacent benign tissue, suggesting that the profiling of THOR methylation can be helpful to stratify cancers with respect to their aggressiveness. Notabily, these results have enable the authors to design Methylation Specific PCR (MSPCR) primers to assay these methylation differences for diagnostic and prognostic purposes.
Based on these results, the author conclude that the methylation profile of the THOR region mirrors the extent of the epigenetic impairment of hTERT expression control in early and advanced urothelial cancer. However the authors didn’t assess whether up-regulation of hTERT mRNA expression in advanced cancers or in transformed cell lines is a direct consequence of the methylation status of the THOR region or the consequence of a general increase of CpG methylation within the entire hTERT promoter. In fact, hyper-methylation of the proximal hTERT promoter (downstream of THOR) has been shown to up-regulate hTERT expression. Therefore, if the authors wish to include their conclusion in the manuscript, my suggestion is to verify it by extending the analysis of methylation profiles to the promoter region downstream of THOR. Otherwise, the authors should change this specific conclusion in the discussion and description of the results, as well as in the abstract.
In conclusion, as already pointed above, the manuscript has the great value of having expanded the methylation analysis of the THOR by defining precisely the most differentially methylated CpGs of THOR in early urothelial cancer. Besides, the optimal design of MSPCR primers allows reliable assays to detect methylation differences for diagnostic and prognostic purposes. However, the authors should respond to the above comments and the following specific comments in order to publish their study in “Cancers”.
Specific comments
1) In Fig.1B and Fig. 3B the authors should explain why different scales were used for the expression of hTAPAS mRNA and hTERT mRNA.
2) Based on the data shown in Fig.3B, the authors conclude that there is an inverse correlation between the levels of TAPAS/TERT expression, without commenting on the low levels of TERT/TAPAS expression in the case of pT3b. Since Fig. 3A shows a different methylation status of THOR in two different pT3b tumors, it would be very useful to know which of the two tumors was used for the data shown in Fig.3B. Actually, TAPAS/TERT expression levels should be evaluated in both pT3b tumors shown in panelA. Conversely, pT2 methylation profiles should be shown. These further analyses may allow to correlate the expression levels of hTERT7hTAPAS with the methylation status of THOR.
Author Response
Dear reviewer, on behalf of all authors I thank you for reviewing our manuscript and your suggestions to improve it.
General comments
In the manuscript by Ott et al, the authors evaluated DNA methylation profiles of the THOR region of the hTERT promoter in various cell lines, as well as in early noninvasive urothelial cancers and muscle-invasive cancers. Regarding cell lines, the data show increased methylation of THOR at specific CpGs from the primary urothelium (pU) to primary renal epithelial cells (hRECs) and the benign urothelial cell line HBLAK. Furthermore, the authors show that methylation further extended to multiple adjacent CpGs in Hela cells and in the SW1710 cell line, while each CpG was methylated in HEK293 cells. Interestingly, analyses of the mRNA expression of hTERT and the long noncoding RNA of hTAPAS in these cell lines suggest that the specific pattern of CpG methylation detected in hREC and HBLAK cells is associated with high expression of hTAPAS mRNA but low expression of hTERT mRNA, corroborating previous studies showing the ability of hTAPAS to trans-repress hTERT mRNA. On the other hand, the expression levels of hTAPAS/hTERT in HEK293 cells suggest that the fully methylated state of THOR in this cell line may be uncoupled from hTAPAS repression and the induction of hTERT expression may be the consequence of a general increase in CpG methylation within the entire hTERT promoter (see below for further criticism).
As regard urothelial cancer, the specific CpG methylation pattern detected in HBLAK cells was also present in both low-grade and high-grade pTa cancers, whereas in high-grade pT1 muscle-invasive cancer methylation affected every CpG ( as in HEK293 cells).
Similar pattern of THOR methylation were also shown for pT3a and two pT3b micro-dissected tumor tissues and one pT3b adjacent benign tissue, suggesting that the profiling of THOR methylation can be helpful to stratify cancers with respect to their aggressiveness. Notabily, these results have enable the authors to design Methylation Specific PCR (MSPCR) primers to assay these methylation differences for diagnostic and prognostic purposes.
Based on these results, the author conclude that the methylation profile of the THOR region mirrors the extent of the epigenetic impairment of hTERT expression control in early and advanced urothelial cancer. However the authors didn’t assess whether up-regulation of hTERT mRNA expression in advanced cancers or in transformed cell lines is a direct consequence of the methylation status of the THOR region or the consequence of a general increase of CpG methylation within the entire hTERT promoter. In fact, hyper-methylation of the proximal hTERT promoter (downstream of THOR) has been shown to up-regulate hTERT expression. Therefore, if the authors wish to include their conclusion in the manuscript, my suggestion is to verify it by extending the analysis of methylation profiles to the promoter region downstream of THOR. Otherwise, the authors should change this specific conclusion in the discussion and description of the results, as well as in the abstract.
In conclusion, as already pointed above, the manuscript has the great value of having expanded the methylation analysis of the THOR by defining precisely the most differentially methylated CpGs of THOR in early urothelial cancer. Besides, the optimal design of MSPCR primers allows reliable assays to detect methylation differences for diagnostic and prognostic purposes. However, the authors should respond to the above comments and the following specific comments in order to publish their study in “Cancers”.
Now in respond to the above comments the following paragraph has been added into the discussion addressing this point and including the reference pointing to methylation of the core promoter:
“Of note, it must always be kept into account that further differential methylated regions contribute to the tumor associated aberrant expression of hTERT. For instance, it has been shown a positive correlation between hypermethylation status of 27 CpG sites within the hTERT promoter core region, hTERT mRNA expression and telomerase activity in a comprehensive investigation of 56 human tumor cell lines, as well as tumor and normal tissues from different organs [31]. Furthermore the results of our study should not distract from the possible case that despite full methylation of THOR, hTAPAS repression may be in some cases uncoupled from this and induction of hTERT expression may be the consequence of a general increase in CpG methylation within the entire hTERT promoter. This is indicated by the full methylated THOR and the expression of hTAPAS and hTERT of HEK293 (Fig. 1B/C).
Specific comments
1) In Fig.1B and Fig. 3B the authors should explain why different scales were used for the expression of hTAPAS mRNA and hTERT mRNA.
We have replaced Fig. 3B the right diagram with one of the same scale as in Fig. 1B
2) Based on the data shown in Fig.3B, the authors conclude that there is an inverse correlation between the levels of TAPAS/TERT expression, without commenting on the low levels of TERT/TAPAS expression in the case of pT3b.
Now we have added the comment: …but less pronounced in the pT3b sample.
Since Fig. 3A shows a different methylation status of THOR in two different pT3b tumors, it would be very useful to know which of the two tumors was used for the data shown in Fig.3B.
Now we have indicated that in the legend. “In addition two samples of primary urothelium and samples of one pTa LG, one pT2, one pT3a (49% methylation) and one pT3b (96% methylation) were used to measure the expression of lncRNA hTAPAS and HTERT gene (B).”
Actually, TAPAS/TERT expression levels should be evaluated in both pT3b tumors shown in panel A. Conversely, pT2 methylation profiles should be shown. These further analyses may allow to correlate the expression levels of hTERT7hTAPAS with the methylation status of THOR.
We have addressed this point as presented above and further studies will be conducted to comprehensively investigate the uncoupled hTAPAS expression from THOR methylation in relation to subsequent promoter methylation in advanced tumors.
Reviewer 3 Report
My main issue related to this paper is related to the statistical approach - the authors should provide more details about the significance of results (e.g. p values) and/or to include symbols (e.g. *) in graphs for showing this. Otherwise, this a well written paper, but still needs minor improvements - especially related to aforementioned subjects.
Author Response
Dear reviewer, on behalf of all authors I thank you for reviewing our manuscript and your suggestions to improve it.
Comments and Suggestions for Authors
My main issue related to this paper is related to the statistical approach - the authors should provide more details about the significance of results (e.g. p values) and/or to include symbols (e.g. *) in graphs for showing this. Otherwise, this a well written paper, but still needs minor improvements - especially related to aforementioned subjects.
Symbols indicating the results of the statistic analysis have been added in the graphs.
Round 2
Reviewer 2 Report
The new version satisfies my criticism and can be published on Cancers